# Chemical Constituents of the Marine-Derived Fungus *Aspergillus* sp. SCS-KFD66

**DOI:** 10.3390/md16120468

**Published:** 2018-11-26

**Authors:** Chang-Liang An, Fan-Dong Kong, Qing-Yun Ma, Qing-Yi Xie, Jing-Zhe Yuan, Li-Man Zhou, Hao-Fu Dai, Zhi-Fang Yu, You-Xing Zhao

**Affiliations:** 1Hainan Key Laboratory for Research and Development of Natural Product from Li Folk Medicine, Institute of Tropical Bioscience and Biotechnology, Chinese Academy of Tropical Agricultural Sciences, Haikou 571101, China; annncl@163.com (C.-L.A.); kongfandong@itbb.org.cn (F.-D.K.); maqingyun@itbb.org.cn (Q.-Y.M.); xieqingyi@itbb.org.cn (Q.-Y.X.); jingzhe1989@yahoo.com (J.-Z.Y.); zhouliman88@126.com (L.-M.Z.); daihaofu@itbb.org.cn (H.-F.D.); 2College of Food Science and Technology, Nanjing Agricultural University, Nanjing 210095, China

**Keywords:** marine-derived fungus, *Aspergillus* sp., secondary metabolites, antibacterial activity

## Abstract

Five new compounds named asperpenes A-C (**1**–**3**), 12,13-dedihydroversiol (**4**), and methyl 6-oxo-3,6-dihydro-2*H*-pyran-4-carboxylate (**5**), along with 10 known compounds (**6**–**15**), were isolated from the fermentation broth of *Aspergillus* sp. SCS-KFD66 associated with a bivalve mollusk, *Sanguinolaria chinensis*, collected from Haikou Bay, China. The structures of the compounds, including the absolute configurations of their stereogenic carbons, were unambiguously determined by spectroscopic data, single-crystal X-ray diffraction analysis, and electronic circular dichroism (ECD) spectral analysis, along with quantum ECD calculations. The growth inhibitory activity of the compounds against four pathogenic bacterial (*Escherichia coli* ATCC 25922, *Staphylococcus aureus* ATCC 6538, *Listeria monocytogenes* ATCC 1911, and *Bacillus subtilis* ATCC 6633), their enzyme inhibitory activities against acetylcholinesterase and α-glucosidase, and their DPPH radical scavenging activity were evaluated.

## 1. Introduction

In the past few decades, natural products have occupied a very important position in modern drug research and development, providing more efficient means for human health care, nutrition, medical care, and other aspects [1]. From 1940 to 2014, 175 new anticancer drugs were approved worldwide, 75% of which came from natural products or their derivatives [2]. Therefore, the study of natural products is of great significance for drug development. Because of the special environmental conditions, marine fungi have been proven to be a rich source of various types of compounds with complex structures and remarkable activities, thereby attracting the attention of for which many natural product chemists turned their attention to them [3,4].

Our previous research on secondary metabolites from marine animal-derived fungi have led to the isolation and identification of a series of structurally new and biologically active natural products, including new quorum-sensing inhibitors from *Penicillium* sp. SCS-KFD08, chlorinated meroterpenoids with anti-H1N1 activity from *Penicillium* sp. SCS-KFD09, and helvolic acid derivatives with potent antibacterial activity from *Aspergillus fumigatus* HNMF0047 [5,6,7,8,9]. In the course of our ongoing research, *Aspergillus* sp. SCS-KFD66 was isolated and identified from a bivalve mollusk, *Sanguinolaria chinensis*, from Haikou Bay, Hainan province, in China. The chemical investigation on the EtOAc extract of the fungal fermentation broth led to the isolation and purification of five new compounds, named asperpenes A-C (**1**-**3**), 12,13-dedihydroversiol (**4**), and methyl 6-oxo-3, 6-dihydro-2*H*-pyran-4-carboxylate (**5**), as well as 10 known compounds, i.e., versiol (**6**) [10], (*E*)-4-oxonon-2-enoic acid (**7**) [11], ergosta-5, 7,22-triene-3*β*-ol (**8**) [12], *β*-sitosterol (**9**) [13], (22*E*)-5α,8α-epidioxyergosta-6,22-dien-3*β*-ol (**10**) [14], 15*α*-hydroxy-(22*E*,24*R*)-ergosta-3,5,8(14),22-tetraen-7-one (**11**) [15], volemolide (**12**) [16], oxaline (**13**) [17], fumitremorgin B (**14**) [18], and helvolic acid (**15**) [19] (Figure 1). Herein, the structure and bioactivities of these compounds are reported.

## 2. Results and Discussion

Compound **1** was obtained as a colorless crystal, and its molecular formula C_15_H_20_O_4_ was established from the HRESIMS *m*/*z* 263.1283 [M − H]^−^. The IR absorptions at 3422, 1695, and 1622 cm^−1^ revealed the presence of a hydroxyl and a conjugated carboxylic group, respectively, which was further confirmed by a characteristic UV *λ*_max_ at 221 nm. The ^1^H and ^13^C NMR spectra (Appendix A) in combination with the HSQC spectra (Appendix A) revealed the presence of two methyls, four sp^3^ methylenes, three sp^3^ methines, two sp^3^ non-protonated carbons, one tri-substituted double bond, and two carboxylic groups. These data were closely related to those of russujaponol H [20], suggesting that **1** was also an illudoid sesquiterpene. The COSY correlations (Appendix A) revealed the connectivities from in CH-1–CH-2–CH-9–CH_2_-10, CH-8–CH-9, and CH_2_-4–CH_2_-5–CH-6. These structure fragments were assembled into a whole structure on the basis of the HMBC correlations (Appendix A) from H_3_-12 (*δ*_H_ 1.25) to C-2 (*δ*_C_ 47.0), C-3 (*δ*_C_ 49.2), C-4 (*δ*_C_ 26.6), and C-6 (*δ*_C_ 37.2), from H_3_-15 (*δ*_H_ 1.18) to C-1 (*δ*_C_ 39.8), C-10 (*δ*_C_ 44.5), C-11 (*δ*_C_ 39.0), and C-14 (*δ*_C_ 182.4), from H-6 (*δ*_H_ 2.54) to C-5 (*δ*_C_ 30.7) and C-7 (*δ*_C_ 133.9), and from H-8 (*δ*_H_ 6.82) to C-13 (*δ*_C_ 170.3) (Figure 2). ROESY correlations from H-6/H-1*β* (*δ*_H_ 0.96)/H_3_-15 and H-8 (*δ*_H_ 6.82)/H-10*β* (*δ*_H_ 1.59) (Figure 3) suggested that H-6 and H_3_-15 were on the same face of the ring system, and H-2 (*δ*_H_ 1.98) and H-9 (*δ*_H_ 2.92) were on the opposite face of the molecule (Figure 3). To support the above deduction and determine the absolute configuration of **1**, a single-crystal X-ray diffraction pattern was obtained using the anomalous scattering of Cu Kα radiation (Figure 4), allowing an explicit assignment of the absolute structure as 2*S*, 3*R*, 6*R*, 9*S*, and 11*R*. This was further corroborated by electronic circular dichroism (ECD) quantum chemical calculations in Gaussian 03 [7]. The experimental and calculated ECD spectra for (2S, 3*R*, 6*R*, 9*S*, 11*R*)-**1** showed good agreement (Figure 5). Thus, **1** was elucidated and named asperpene A.

Compound **2** was obtained as a colorless powder, whose molecular formula was established as C_15_H_22_O_3_ by HRESIMS *m*/*z* 273.1455 [M + Na]^+^. Comparison of the ^1^H and ^13^C NMR data (Appendix A) of **2** with those of **1** revealed the presence of a hydroxymethyl group (*δ*_C/H_ 72.1/3.3, C-14) and a carboxylic group in **2** instead of two carboxyl groups as in **1**. The above data, together with HMBC correlations from H_3_-15 (*δ*_H_ 0.96) to the hydroxymethyl carbon (*δ*_C_ 72.1), indicated that the carboxylic group in **1** was replaced by a hydroxymethyl in **2**. In the ROESY spectrum (Figure 3), correlations from H_2_-14 (*δ*_H_ 3.31, 3.33)/H-1α (*δ*_H_ 1.56), H-6 (*δ*_H_ 1.91)/H-1*β* (*δ*_H_ 0.85), and H-9 (*δ*_H_ 2.88)/H-4α (*δ*_H_ 1.86) suggested that **2** shared the same configuration at the stereogenic C-2, C-3, C-6, C-9, and C-11. Thus, the structure of **2** was established and named as asperpene B.

Compound **3** possessed the same molecular formula as **2,** as determined by HRESIMS data. The ^1^H and ^13^C NMR data of **3** (Appendix A) were also quite similar to those of **2**. However, in the HMBC spectrum of **3** (Figure 2), correlations from H_3_-15 (*δ*_H_ 1.26) to the carbonyl (*δ*_C_ 201.5) and from H_2_-13 (*δ*_H_ 4.01, 4.03) to C-7 (*δ*_C_ 139.5) and C-8 (*δ*_C_ 125.5) suggested the positions of the carboxylic acid and the hydroxymethyl groups at C-11 and C-7, respectively, which is resulted different from those of **2**. ROESY correlations (Figure 3) of H-10*β* (*δ*_H_ 1.57)/H-8 (*δ*_H_ 5.49) and H_3_-15/H-1*β* (*δ*_H_ 1.09)/H-6 (*δ*_H_ 1.99) suggested that **3** had the same configurations of its stereogenic carbons as **2**.

Compound **4** was obtained as a yellow oil, and its molecular formula was determined as C_16_H_20_O_3_ on the basis of the HRESIMS data, implying seven degrees of unsaturation. The ^1^H and ^13^C NMR data (Appendix A) of **4** indicated the presence of three methyls, one methylene, eight methines (including five olefinic and one oxygenated), and four non-protonated carbons (including one ketone carbonyl, one olefinic, and one oxygenated sp^3^). These data were quite similar to those of versiol (**6**) [10], suggesting that they were structurally related, and the only difference was that one disubstituted double bond in **4** was saturated to two vicinal methylenes in **6**, as supported by HMBC correlations from H-13 (*δ*_H_ 7.22) to C-8 (*δ*_C_ 85.9) and from H-12 (*δ*_H_ 5.46) to C-11 (*δ*_C_ 198.8). ROESY correlation between H-2*β* (*δ*_H_ 1.19) and H-10 (*δ*_H_ 2.69) suggested their cofacial relationship, while the absence of ROESY correlations from H_3_-14 (*δ*_H_ 1.46) and H_3_-15 (*δ*_H_ 1.17) to H-10 suggested that H_3_-14 and H_3_-15 were on the opposite face with respect to H-10. Considering that versiol (**6**) and **4** were biosynthetically related and a relatively large amount of versiol (**6**) was isolated, we crystalized versiol (**6**) successfully and subjected it to a single-crystal X-ray diffraction experiment (Figure 4), finally allowing an explicit assignment of the absolute structure of versiol (**6**) as 1*S*, 3*S*, 8*S*, 9*R*, and 10*S*. The absolute configurations of the stereogenic carbons of **4** were also suggested to be 1*S*, 3*S*, 8*S*, 9*R*, and 10*S* on the basis of a biosynthetic consideration. The experimental ECD spectrum (Figure 5) of **4** showed characteristic exciton CDs absorption bands at 261 (-0.31) and 230 (+0.14) nm due to the a negative couplet of the *α*,*β*-unsaturated carbonyl and the conjugated double-bond moieties, which further confirmed the absolute configuration assignment. Moreover, the experimental and calculated ECD spectra for **4** also matched well (Figure 5).

Compound **5** was isolated as a colorless oil, whose molecular formula was established as C_7_H_8_O_4_ by HRESIMS *m*/*z* 179.0316 [M + Na]^+^. The ^1^H, ^13^C, and HSQC NMR spectra (Appendix A) of **5** showed signals for two ester carbonyls (*δ*_C_ 165.0, 163.6), one tri-substituted double bond (*δ*_C/H_ 126.1/6.77, 145.3), two sp^3^ methylenes, one of which is oxgenated (*δ*_C/H_ 66.7/4.46), and one methoxyl group (*δ*_C/H_ 53.1/3.87). COSY correlations (Appendix A) of H_2_-2 (*δ*_H_ 4.46)/H_2_-3 (*δ*_H_ 2.71) and HMBC correlations (Appendix A) from H_2_-2 (*δ*_H_ 4.46) and H-5 (*δ*_H_ 6.77) to C-6 (*δ*_C_ 163.6) and C-4 (*δ*_C_ 145.3) and from H_2_-3 and H_3_-8 (*δ*_H_ 3.87) to C-7 (*δ*_C_ 165.0) led to the determination of the full structure of **5,** as shown in Figure 1.

Compounds **1**–**15** were tested for their antibacterial activity against *Escherichia coli* ATCC 25922, *Staphylococcus aureus* ATCC 6538, *Listeria monocytogenes* ATCC 1911, and *Bacillus subtilis* ATCC 6633 by the 96-well microtiter plates method [21]. The results (Table 1) revealed that **7**, **8**, **12**, and **13** showed inhibitory activities against *B. subtilis* ATCC 6633, with MIC values of 4, 128, 128, and 128 μg/mL, respectively, whereas **7**, **8**, **14**, and **15** showed inhibitory activity against *S. aureus* ATCC 6538, with MIC values of 16, 128, 128 and 2 μg/mL, respectively; **15** also had inhibitory activity against *L. monocytogenes* ATCC 1911, with MIC value of 128 μg/mL. None of these compounds showed inhibitory activity against *E. coli* ATCC 25922.

Additionally, the DPPH radical scavenging activity and the acetylcholinesterase and α-glucosidase inhibitory activities of all the isolated compounds were evaluated by the DPPH method [22], Ellman colorimetric method [23], and PNPG method [24], respectively. None of these compounds showed inhibitory activities against α-glucosidase and acetylcholinesterase. However, **1**, **3**, **4**, **8**, **11**, and **15** showed weak DPPH radical scavenging activity, with IC_50_ values of 1.8, 0.6, 1.1, 0.6, 1.2, and 0.7 mM (ascorbic acid as positive control, IC_50_ 0.04 mM).

## 3. Experimental Section

### 3.1. General Experimental Procedure

Optical rotations were measured with a JASCO P-1020 digital polarimeter. The IR spectra were obtained on with a Nicolet Nexus 470 spectrophotometer as KBr discs. The UV spectra were obtained from with a Beckman DU 640 spectrophotometer. ECD data were measured collected on using a JASCO J-715 spectropolarimeter. The NMR spectra were recorded on a Bruker AV-500 spectrometer with TMS as an internal standard. ESIMS, HRESIMS, and HREIMS data were acquired on a Micromass Autospec-Ultima-TOF, API QSTAR Pulsar 1, or Waters Autospec Premier spectrometer. The sea salt was produced by evaporation of seawater collected in Laizhou Bay, Weifang, China (Weifang HaiHua Yu Feng Chemical Factory). Semi-preparative HPLC separation was used octadecyl silane (ODS) columns (YMC-pack ODS-A, 10 × 250 mm, 5 μm, 4 mL/min) and Ph column (YMC-pack Ph, 10 × 250 mm, 5 μm, 4 mL/min) for separation. Thin-layer chromatography (TLC) and column chromatography (CC) were carried out on precoated silica gel GF_254_ (10–40 μm, Qingdao Marine Chemical Inc., Qingdao, China) and silica gel (200–300 mesh, Qingdao Marine Chemical Inc., Qingdao, China), respectively.

### 3.2. Fungal Material and Fermentation

The strain SCS-KFD66 was isolated from a bivalve mollusk, Sanguinolaria chinensis, collected from Haikou Bay, Hainan province, in China. After grinding, the sample (1 g) was diluted to 10^−2^ g/mL with sterile H_2_O, 100 μL of which was spread on a PDA (200 g potato, 20 g glucose, 20 g agar per liter of sea water collected in Haikou Bay, China) plate containing chloramphenicol (100 μg/mL) as a bacterial inhibitor. Fungal identification was carried out by its examining the morphological characteristics and 18S rRNA gene sequences (GenBank accession No. MK085984, Appendix A) with of the single coloniesy. A reference culture of Aspergillus sp. SCS-KFD66 is deposited in our laboratory and which maintained at −80 °C. The isolate was cultured on slants of PDA medium at 28 °C for 5 days and then transferred to two hundred 1 L Erlenmeyer flasks containing solid rice medium (80 g rice, 3.96 g sea salt, 120 mL tap water, pH 7.0), used for fermentation. The flasks were incubated under static conditions at room temperature for 30 days. 

### 3.3. Extraction and Isolation

The fermented cultures were extracted with three-fold volumes (3 × 300 mL) of EtOAc, then filtered through a cheesecloth to separate the rice from the mixture. After repeating the procedure three times, the EtOAc extracts were evaporated under a reduced pressure to produce 409.6 g of a crude extract. The extract was fractionated by a silica gel VLC column using different solvents of increasing polarity, from petroleum ether to EtOAc, to yield seven fractions (Frs. 1−7). Fr. 3 (5.3 g) was further purified by HPLC using an octadecyl silane (ODS) silica gel column and eluted with in a MeOH/H_2_O (1:5, 2:3, 3:2, 4:1, 1:0) gradient to afford **8** (3.5 mg), **9** (14.0 mg), and three subfractions (Sfrs. 3-1–Fr. 3-3). Sfr. 3-2 (289.7 mg) was subjected to VLC on silica gel and eluted with an EtOAc/petroleum ether stepwise gradient (from 1:10 to 2:1) to afford **4** (3.6 mg). Fr. 4 (8.0 g) was separated into seven subfractions (Sfrs. 4-1–Fr–4-7) by HPLC using an ODS silica gel column with a gradient elution of MeOH/H_2_O (1:5, 2:3, 3:2, 4:1, 1:0). Sfr. 4-1 (16.5 mg) was purified by a semipreparative HPLC (YMC-pack ODS-A, 5 μm; 10 × 250 mm; 35% MeCN/H2O; containing 0.1% TFA; 4 mL/min) to afford **7** (*t*_R_ 18.4 min; 1.7 mg). Fr. 5 (3.7 g) was chromatographed on an ODS silica gel column with a gradient elution of MeOH/H_2_O (1:5, 2:3, 3:2, 4:1, 1:0) to yield **14** (3.0 mg), **15** (4.1 mg), **10** (5.2 mg), and four subfractions (Sfrs. 5-1–Fr. 5-4). Sfr. 5-4 (13.3 mg) was purified by a semipreparative HPLC (YMC-pack ODS-A, 5 μm; 10 × 250 mm; 40% MeCN/H_2_O; containing 0.1% TFA; 4 mL/min) to afford **6** (*t*_R_ 9.8 min; 6.0 mg). Fr. 6 (4.0 g) was fractionated on an ODS silica gel column with a gradient elution of MeOH/H_2_O (1:5, 2:3, 3:2, 4:1, 1:0) to yield six subfractions (Sfrs. 6-1–Fr. 6-6). Sfr. 6-4 (69.9 mg) was subjected to semipreparative HPLC (YMC-pack Ph, 5 μm; 10 × 250 mm; 30% MeCN/H_2_O; containing 0.1% TFA; 4 mL/min) to afford **1** (*t*_R_ 15.6 min; 24.5 mg), **2** (*t*_R_ 18.4 min; 2.5 mg), and **3** (*t*_R_ 21.5 min; 3.7 mg). Purification of Fr. 7 (28.3 g) by a silica gel VLC column with a stepwise gradient with of MeOH/CHCl_3_ (from 10:90 to 100:0) gave **13** (78.5 mg) and eight fractions (Sfrs. 7-1–Fr. 7-8). Sfr. 7-1 (119.2 mg) was applied to ODS silica gel with a gradient elution of MeOH/H_2_O (1:5, 2:3, 3:2, 4:1, 1:0) to yield **12** (2.0 mg). Sfr. 7-2 (199.4 mg) was purified by ODS silica gel column with a gradient elution with of MeOH/H_2_O (1:5, 2:3, 3:2, 4:1, 1:0) to give **11** (3.0 mg). Sfr. 7-4 (184.9 mg) was purified by Sephadex LH-20 chromatography and eluted with MeOH to give three subfractions (Sfrs. 7-4-1–Fr. 7-4-3). Sfr. 7-4-1 (149.6 mg) was finally purified by semipreparative HPLC (YMC-pack ODS-A, 5 μm; 10 × 250 mm; 40% MeOH/H_2_O; 4 mL/min) to obtain **5** (*t*_R_ 4.4 min; 9.3 mg).

Asperpene A (**1**): Colorless crystal; mp 194–195 °C; ${\left[\alpha \right]}_{D}^{25}$ +8 (*c* 0.1, MeOH); UV (MeOH) *λ*_max_ (log *ε*): 203 (3.54) nm; ECD (MeOH) *λ*_max_ 221 (+0.34) nm; IR (KBr) *ν*_max_ (cm^−1^): 3422, 2930, 2851, 1695, 1626, 1453, 1390, 1254, 1121. ^1^H NMR data, Table 2; ^13^C NMR data, Table 3; HRESIMS *m*/*z* 263.1283 [M − H]^−^ (calcd for C_15_H_19_O_4_, 263.1289).

Asperpene B (**2**): White powders; ${\left[\alpha \right]}_{D}^{25}$ −6 (*c* 0.1, MeOH); UV (MeOH) *λ*_max_ (log *ε*): 213 (3.14) nm, 215 (3.14) nm; ECD (MeOH) *λ*_max_ 295 (-0.06), 255 (+0.02), 229 (-0.08) nm; IR (KBr) *ν*_max_ (cm^−1^): 3446, 2932, 2867, 1692, 1638, 1455, 1393, 1255, 1049. ^1^H NMR data, Table 2; ^13^C NMR data, Table 3; HRESIMS *m*/*z* 273.1455 [M + Na]^+^ (calcd for C_15_H_23_O_3_Na, 273.1461).

Asperpene C (**3**): White powders; ${\left[\alpha \right]}_{D}^{25}$ +5 (*c* 0.1, MeOH); UV (MeOH) *λ*_max_ (log *ε*): 202 (3.34) nm; IR (KBr) *ν*_max_ (cm^−1^): 3415, 2959, 1708, 1456, 1184, 1122. ^1^H NMR data, Table 2; ^13^C NMR data, Table 3; HRESIMS *m*/*z* 273.1457 [M + Na]^+^ (calcd for C_15_H_22_O_3_Na, 273.1461).

12,13-Dedihydroversiol (**4**): Yellow oil; ${\left[\alpha \right]}_{D}^{25}$ −8 (*c* 0.1, MeOH); UV (MeOH) *λ*_max_ (log *ε*): 260 (3.46) nm, 234 (3.58) nm, 219 (3.66) nm; ECD (MeOH) *λ*_max_ 310 (+0.02), 261 (-0.31), 230 (+0.14) nm; IR (KBr) *ν*_max_ (cm^−1^): 3445, 2930, 1723, 1655, 1605, 1454, 1385, 1266, 1108. ^1^H NMR data, Table 2; ^13^C NMR data, Table 3; HRESIMS *m*/*z* 283.1297 [M + Na]^+^ (calcd for C_16_H_20_O_3_Na, 283.1305).

Methyl 6-oxo-3, 6-dihydro-2*H*-pyran-4-carboxylate (**5**): Colorless oil; UV (MeOH) *λ*_max_ (log *ε*): 218 (3.45); IR (KBr) *ν*_max_ (cm^−1^): 2927, 1726, 1641, 1441, 1260, 1219, 1084. ^1^H NMR data, Table 2; ^13^C NMR data, Table 3; HRESIMS *m*/*z* 179.0316 [M + Na]^+^ (calcd for C_7_H_8_O_4_Na, 179.0315).

**X-ray Crystal Data for 1** and **6**: Colorless crystals of **1** and **6** were obtained in the mixed solvent of MeOH and H_2_O. Crystal data of **1** and **6** were obtained on a Bruker D8 QUEST diffractometer (Bruker) with graphite monochromated Cu Kα radiation (λ = 1.54178 Å). Crystallographic data for **1** and **6** have been deposited with in the Cambridge Crystallographic Data Center as supplementary publication numbers CCDC 1875828 and 1875827. These data can be obtained free of charge from The Cambridge Crystallographic Data Centre via www.ccdc.cam.ac.uk/data_request/cif.

Crystal data for **1**. Orthorhombic, C_15_H_20_O_4_; space group P 21 21 21 with a = 7.2684(6) Å, b = 9.6858(7) Å, c = 19.5962(15) Å, *V* = 1379.58(18) Å^3^, *Z* = 4, *D*_calcd_ = 1.273 g/cm^3^, *μ* = 0.747 mm^−1^, and *F*(000) = 568. *T* = 296(2) K. R1 = 0.0583 (I > 2σ(I)), wR2 = 0.1409 (all data), S = 0.978. Absolute structure parameter: 0.0 (4). The structures were solved using ShelXS. The structural solutions were found by direct methods and refined using the ShelXL package by least-squares minimization. The final structures were examined using the Addsym subroutine of PLATON to assure that no additional symmetry could be applied to the models. All non-hydrogen atoms were refined with anisotropic thermal factors.

Crystal data for **6**. Orthorhombic, C_16_H_22_O_3_; space group P 21 21 21 with a = 6.0879(2) Å, b = 9.1456(3) Å, c = 25.1933(9) Å, *V* = 1402.70(8) Å^3^, *Z* = 4, *D*_calcd_ = 1.237 Mg/m^3^, *μ* = 0.674 mm^−1^, and *F*(000) = 564. *T* = 296(2) K. R1 = 0.0339 (I > 2σ(I)), wR2 = 0.0793 (all data), S = 1.060. Absolute structure parameter: 0.08(12). The structures were solved using ShelXS. The structural solutions were found by direct methods and refined using the ShelXL package by least-squares minimization. The final structures were examined using the Addsym subroutine of PLATON to assure that no additional symmetry could be applied to the models. All non-hydrogen atoms were refined with anisotropic thermal factors.

## 4. Conclusions

In conclusion, five new compounds (**1**–**5**) and 10 known compounds (**6**–**15**) were isolated from the fermentation broth of *Aspergillus* sp. SCS-KFD66 which was isolated from a bivalve mollusk, *S.anguinolaria chinensis*, collected from Haikou Bay, China. The structures of the isolated compounds were unambiguously determined by spectroscopic data, single-crystal X-ray diffraction analysis, and comparison of the calculated and experimental ECD spectra. Compounds **7**, **8**, **12**, and **13** showed antibacterial activity against *Bacillus subtilis*, with MIC values of 4, 128, 128, and 128 μg/mL. Compounds **7**, **8**, **14**, and **15** exhibited antibacterial activity against *S.taphylococcus aureus*, with MIC values of 16, 128, 128, and 2 μg/mL, while **15** also showed inhibitory activity against *L.isteria monocytogenes*, with MIC value of 128 μg/mL. Compounds **1**, **3**, **4**, **8**, **11**, and **15** showed a weak DPPH radical scavenging activity, with IC_50_ values of 1.8, 0.6, 1.1, 0.6, 1.2, and 0.7 mM (ascorbic acid as positive control, IC_50_ 0.04 mM).

## Figures and Tables

**Figure 1 marinedrugs-16-00468-f001:**
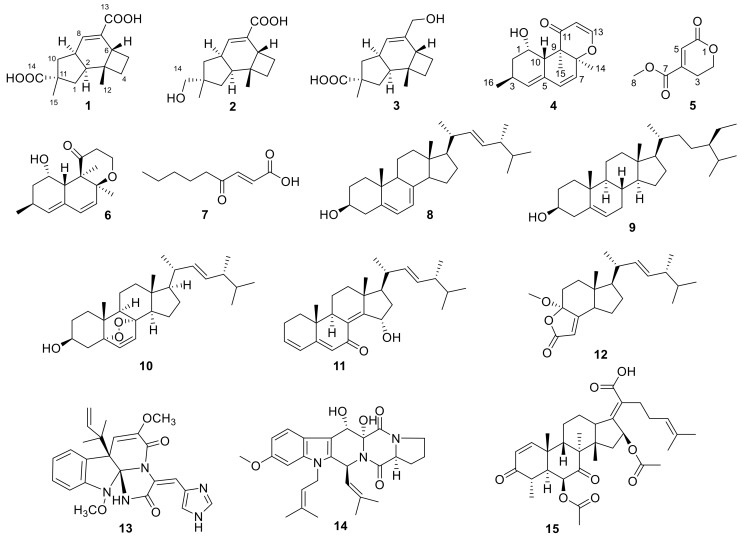
Structures of compounds **1**–**15**.

**Figure 2 marinedrugs-16-00468-f002:**
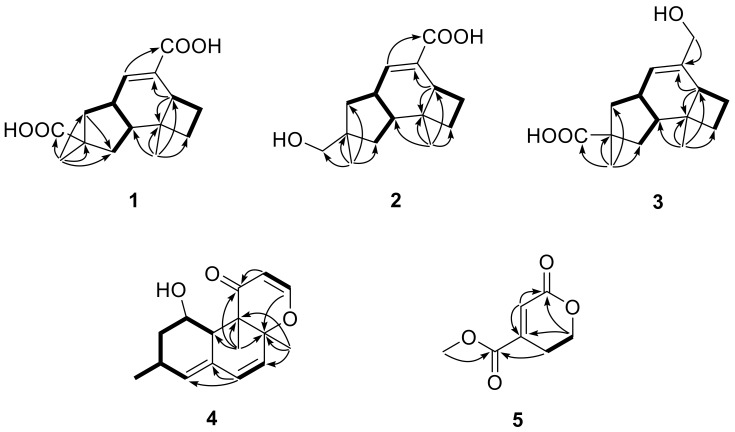
Key COSY (▬) and HMBC (→) correlations of **1**–**5**.

**Figure 3 marinedrugs-16-00468-f003:**
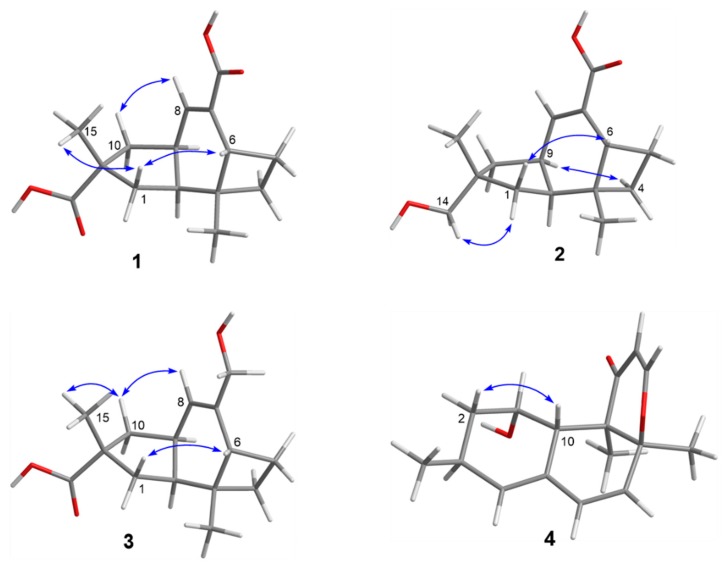
Key ROESY correlations of **1**–**4**.

**Figure 4 marinedrugs-16-00468-f004:**
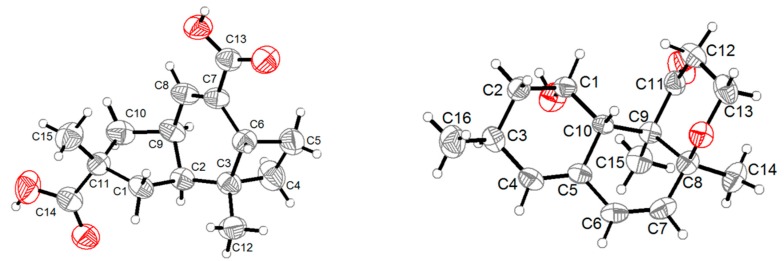
ORTEP diagrams of **1** and **6**.

**Figure 5 marinedrugs-16-00468-f005:**
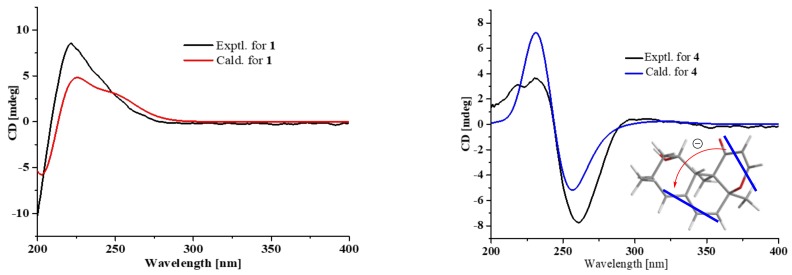
Comparison of measured and calculated ECD spectra for **1** and **4** and ECD exciton chirality model for **4**.

**Table 1 marinedrugs-16-00468-t001:** Antibacterial activities of compounds **7**, **8**, and **12**–**15**.

Compound	MIC (μg/mL)		
	*Staphylococcus aureus* ATCC 6538	*Listeria monocytogenes* ATCC 1911	*Bacillus subtilis* ATCC 6633
**7**	16	>128	4
**8**	128	>128	128
**12**	>128	>128	128
**13**	>128	>128	128
**14**	128	>128	>128
**15**	2	128	>128
Ampicillin ^a^	<1	<1	<1

^a^ Positive control.

**Table 2 marinedrugs-16-00468-t002:** ^1^H NMR data (500 MHz, *δ* in ppm, *J* in Hz) of **1–5.**

Position	1 ^a^	2 ^b^	3 ^b^	4 ^b^	5 ^b^
1	2.11, dd (12.9, 7.1)	1.56, dd (12.8, 7.5)	2.12, dd (12.8, 7.3)	3.95, m	
	0.96, dd (12.9, 12.1)	0.85, dd (12.8, 12.8)	1.09, dd (12.7, 12.7)		
2	1.98, m	2.60, m	2.24, m	1.96, m	4.46, t (6.2)
				1.19, ddd (12.6, 12.6, 1.7)	
3				2.64, m	2.71, td (6.2, 1.7)
4	2.49, m	1.86, m	1.96, m	5.81, d (2.0)	
	1.42, m	1.48, m	1.48, m		
5	1.90, m	2.51, m	2.41, m		6.77, t (1.7)
	1.46, m	1.48, m	1.48, m		
6	2.54, m	1.91, overlap	1.99, m	6.31, d (9.6)	
7				5.48, d (9.6)	
8	6.82, d (2.3)	6.97, d (2.2)	5.49, d (2.0)		3.87, s
9	2.92, m	2.88, m	2.81, m		
10	2.66, dd (13.6, 8.9)	1.95, dd (13.7, 8.9)	2.59, dd (13.5, 8.2)	2.69, m	
	1.59, dd (13.6, 1.9)	1.50, overlap	1.57, dd (13.5, 1.6)		
12	1.25, s	1.24, s	1.22, s	5.46, d (6.0)	
13			4.01, br d (14.3)	7.22, d (6.0)	
			4.03, br d (14.3)		
14		3.31, d (16.5)		1.46, s	
		3.33, d (16.5)			
15	1.18, s	0.96, s	1.26, s	1.17, s	
16				1.04, d (7.2)	

^a^ Taken in CD_3_OD, ^b^ Taken in CDCl_3_.

**Table 3 marinedrugs-16-00468-t003:** ^13^C NMR data (125 MHz, *δ *in ppm) of **1–5**.

Position	1 ^a^	2 ^b^	3 ^b^	4 ^b^	5 ^b^
	*δ*_C,_ Type	*δ*_C,_ Type	*δ*_C,_ Type	*δ*_C,_ Type	*δ*_C,_ Type
1	39.8, CH_2_	36.7, CH_2_	38.9, CH_2_	66.8, CH	
2	47.0, CH	35.8, CH	36.7 CH	38.4, CH_2_	66.7, CH_2_
3	49.2, C	38.3, C	38.2, C	25.4, CH	23.7, CH_2_
4	26.6, CH_2_	30.0, CH_2_	30.4, CH_2_	138.2, CH	145.3, C
5	30.7, CH_2_	25.7, CH_2_	24.7, CH_2_	129.6, C	126.1, CH
6	37.2, CH	45.4, CH	46.6, CH	134.9, CH	163.6, C
7	133.9, C	132.0, C	139.5, C	124.9, CH	165.0, C
8	143.3, CH	145.1, CH	125.5, CH	85.9, C	53.1, CH_3_
9	40.8, CH	40.2, CH	38.8, CH	50.9, C	
10	44.5, CH_2_	42.2, CH_2_	44.3, CH_2_	40.8, CH	
11	39.0, C	42.9, C	48.3, C	198.8, C	
12	26.5, CH_3_	26.2, CH_3_	27.5, CH_3_	104.6, CH	
13	170.3, C	171.2, C	65.5, CH_2_	161.0, CH	
14	182.4, C	72.1, CH_2_	184.7, C	19.2, CH_3_	
15	27.8, CH_3_	26.8, C	26.3, CH_3_	13.5, CH_3_	
16				21.2, CH_3_	

^a^ Taken in CD_3_OD, ^b^ Taken in CDCl_3_.

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
