# Peer review of "Chemical Constituents of the Marine-Derived Fungus Aspergillus sp. SCS-KFD66"

_marinedrugs, 2018, doi:10.3390/md16120468_

Reviewer 1 Report

An et al present an article on the characterization of compounds isolated from an Aspergillus species associated with Sanguinolaria chinensis. In general the article fits the criteria for consideration in the journal Marine Drugs, contains interesting information and is briefly and concisely written.

However the article should be reevaluated after considering specific points.

The information given in the Introduction could well be expanded and include a few more details from previous studies.

Authors present a supplement containing 39 additional figures. These figures are not cited at all in the text. Please include this and also other relevant information in the main manuscript and the relevant figures.

According to the “List of Supporting Information” (Supplement/Page 2) and the title of Supplement/Page 4, a phylogenetic tree of Aspergillus sp. SCS-KFD66 is presented. However no such figure appears in the Supplement.

In Chapter 3.2, authors state that the 18S rRNA gene sequence was deposited in GenBank with accession No. MK085984. No results appear in GenBank under this number. Please verify and confirm the number given in the text.

Other Points

The antibacterial activity of these compounds against bacterial species and also enzyme inhibition assays are presented. To make it reader friendlier, authors may want to consider presenting part of these data in the form of a Figure, or a Table.

Authors may want to consider modifying the name “aspergoid”, as it also used in correlation with the Asperger syndrome.

Author Response

Author's Reply to the Review Report (Reviewer 1)

1.The information given in the Introduction could well be expanded and include a few more details from previous studies.

R: The introduction has been revised as suggested.

2.Authors present a supplement containing 39 additional figures. These figures are not cited at all in the text. Please include this and also other relevant information in the main manuscript and the relevant figures.

R: The additional 39 figures had been included in the Supplementary Materials section in the manuscript.

3.According to the “List of Supporting Information” (Supplement/Page 2) and the title of Supplement/Page 4, a phylogenetic tree of Aspergillus sp. SCS-KFD66 is presented. However no such figure appears in the Supplement.

R: Sorry for our mistake. We have deleted the description of “the phylogenetic tree”.

4. In Chapter 3.2, authors state that the 18S rRNA gene sequence was deposited in GenBank with accession No. MK085984. No results appear in GenBank under this number. Please verify and confirm the number given in the text.

R: We have verify and confirm accession No. MK085984 given in the text. The number is correct and it has not been released by GenBank.

5.The antibacterial activity of these compounds against bacterial species and also enzyme inhibition assays are presented. To make it reader friendlier, authors may want to consider presenting part of these data in the form of a Figure, or a Table.

R: We have presented antibacterial activities in the form of a Table as suggested.

6.Authors may want to consider modifying the name “aspergoid”, as it also used in correlation with the Asperger syndrome.

R: The name “aspergoid” has been changed to asperpene as suggested.

Reviewer 2 Report

This manuscript presents the structure elucidation and bioactivity of five new compounds isolated from Aspergillussp. The work seems carefully done and the introduction, experiment design, analyses, results, and conclusions are clearly presented, except for MS spectra of compounds 1-3, and 5. If HRMS were measured with a narrow mass range, LRMS with a wide mass range should be also provided. Since the MS of compound 1 provided in Supplementary Materials seems to be a calculated one, the measured MS spectrum should be provided.

The manuscript needs corrections as follows.

(1) Page 2, line 52: ‘[M–H]+’ should be corrected to [M–H]–’.

(2) Page 2, line 61: ‘from H-6 to H-7’ should be corrected to ‘from H-6 to C-5 and C-7’.

(3) Page 2, line 82: ‘enter’ should be corrected to ‘center’.

(4) Page 4, line 114: ‘methlenes’ should be corrected to ‘methylenes’.

(5) Page 7, lines 174, 177-178, 182, 188-189, and 190: ‘octadecyl silane (ODS)’ should be corrected toODS’.

(6) Page 7, line 187: ‘chloroform ether’ should be corrected.

(7) Page 7, line 197: ‘[M–H]+’ should be corrected to [M–H]–’.

(8) Page 7, line 198: ‘C15H20O4’ should be corrected to C15H19O4’.

(9) Page 7, line 202: ‘C15H22O3’ should be corrected to C15H23O3’.

(10) Page 8, line 205: ‘C15H22O3’ should be corrected to C15H22O3Na’.

(11) Page 8, line 209: ‘C16H20O3’ should be corrected to C16H20O3Na’.

(12) Page 8, line 212: ‘C7H8O4’ should be corrected to C7H8O4Na’.

Author Response

Author's Reply to the Review Report (Reviewer 2)

1. This manuscript presents the structure elucidation and bioactivity of five new compounds isolated from Aspergillussp. The work seems carefully done and the introduction, experiment design, analyses, results, and conclusions are clearly presented, except for MS spectra of compounds 1-3, and 5. If HRMS were measured with a narrow mass range, LRMS with a wide mass range should be also provided. Since the MS of compound 1 provided in Supplementary Materials seems to be a calculated one, the measured MS spectrum should be provided.

R: We have processed the HRMS spectra in the Supplementary Materials as suggested.

2. ‘[M–H]+’ should be corrected to ‘[M–H]–’.

R: ‘[M–H]+’ has been revised to to ‘[M–H]–’.

3.Page 2, line 61: ‘from H-6 to H-7’ should be corrected to ‘from H-6 to C-5 and C-7’.

R: ‘From H-6 to H-7’ has been corrected to ‘from H-6 to C-5 and C-7’.

4.Page 2, line 82: ‘enter’ should be corrected to ‘center’.

R: The sentence has been corrected.

5. Page 4, line 114: ‘methlenes’ should be corrected to ‘methylenes’.

R: ‘methlenes’ has been revised to ‘methylenes’.

6.Page 7, lines 174, 177-178, 182, 188-189, and 190: ‘octadecyl silane (ODS)’ should be corrected to‘ODS’.

R: ‘octadecyl silane (ODS)’ has been corrected to‘ODS’.

7. Page 7, line 187: ‘chloroform ether’ should be corrected.

R: ‘chloroform ether’ has been corrected to ‘chloroform’

8.Page 7, line 197: ‘[M–H]+’ should be corrected to ‘[M–H]–’.

R: ‘[M–H]+’ has been corrected to ‘[M–H]–’

9.Page 7, line 198: ‘C15H20O4’ should be corrected to ‘C15H19O4’.

R: ‘C15H20O4’ has been corrected to ‘C15H19O4’.

10. Page 7, line 202: ‘C15H22O3’ should be corrected to ‘C15H23O3’.

R: ‘C15H22O3’ has been corrected to ‘C15H23O3’.

11.Page 8, line 205: ‘C15H22O3’ should be corrected to ‘C15H22O3Na’.

R: ‘C15H22O3’ has been corrected to ‘C15H22O3Na’.

12.Page 8, line 209: ‘C16H20O3’ should be corrected to ‘C16H20O3Na’.

R: ‘C16H20O3’ has been corrected to ‘C16H20O3Na’.

13.Page 8, line 212: ‘C7H8O4’ should be corrected to ‘C7H8O4Na’.

R: ‘C7H8O4’ has been corrected to ‘C7H8O4Na’.

Round  2

Reviewer 1 Report

No further comments

Author Response

Dear reviewer, 

Thank you very much for your comments on our manuscript. According to these suggestions, we have checked and revised the manuscript carefully. 

Thank you very much for your kind helps to our work again.